# Cross-sectional study of the association of social relationship resources with *Staphylococcus aureus* colonization in naturally occurring social groups along the US/Mexico border

Steven D. Barger[1]*, Monica R. Lininger[2], Robert T. Trotter, II[3,4], Mimi Mbegbu[5], Shari Kyman[5], Kara Tucker-Morgan[6], Colin Wood[5], Briana Coyne[5], Benjamin Russakoff[5], Kathya Ceniceros[6], Cristina Padilla[6], Sara Maltinsky[5], Talima Pearson[3,5]*

1 Department of Psychological Sciences, Northern Arizona University, Flagstaff, AZ, United States of America, 2 Department of Physical Therapy and Athletic Training, Northern Arizona University, Flagstaff, AZ, United States of America, 3 Center for Health Equity Research, Northern Arizona University, Flagstaff, AZ, United States of America, 4 Department of Anthropology, Northern Arizona University, Flagstaff, AZ, United States of America, 5 Pathogen & Microbiome Institute, Northern Arizona University, Flagstaff, AZ, United States of America, 6 Northern Arizona University, Yuma, Arizona, United States of America

* talima.pearson@nau.edu (TP); steven.barger@nau.edu (SDB)

## Abstract

Asymptomatic carriage of *Staphylococcus aureus* is a major risk factor for subsequent clinical infection. Diminishing returns from mitigation efforts emphasize the need to better understand colonization, spread, and transmission of this opportunistic pathogen. While contact with other people presents opportunities for pathogen exposure and transmission, diversity of social connections may be protective against pathogens such as the common cold. This study examined whether social relationship resources, including the amount and diversity of social contacts, are associated with *S. aureus* colonization. Participants were community members (N = 443; 68% Hispanic) in naturally occurring social groups in southwestern Arizona. Four types of social relationships and loneliness were assessed, and samples from the skin, nose and throat were obtained to ascertain *S. aureus* colonization. Overall *S. aureus* prevalence was 64.8%. Neither the amount nor the diversity of social contacts were associated with *S. aureus* colonization. The concurrent validity of the social relationship assessments was supported by their moderate intercorrelations and by their positive association with self-rated health. The results suggest that the association of social network diversity and susceptibility to the common cold does not extend to *S. aureus* colonization. Conversely, colonization prevalence was not higher among those with more social contacts. The latter pattern suggests that social transmission may be relatively infrequent or that more intimate forms of social interaction may drive transmission and colonization resulting in high community prevalence of *S. aureus* colonization. These data inform communicable disease control efforts.

**Data Availability Statement:** All relevant data are within the paper and its Supporting Information files.

**Funding:** This work was supported by the NAU Southwest Health Equity Research Collaborative, which is funded by the National Institute on Minority Health and Health Disparities at the National Institutes of Health (grant NIH U54MD012388). Funding was also received from the National Institute of Allergy and Infectious Diseases at the National Institutes of Health (R15AI156771) The funders had no role in study design, data collection and analysis, decision to publish, or preparation of the manuscript.

**Competing interests:** The authors have declared that no competing interests exist.

**Abbreviations:** MRSA, methicillin-resistant *Staphylococcus aureus*; MSSA, methicillin-susceptible *Staphylococcus aureus*; SNI, social network index; SRH, self-rated health; NHIS, National Health Interview Survey; PCR, polymerase chain reaction; DNA, deoxyribonucleic acid.

## Introduction

Social network diversity, denoted by the number of different social relationship roles in which one participates (i.e., spouse; parent; work; school; neighbor), presents opportunities for pathogen exposure and subsequent transmission. Conversely, social network diversity is associated with a lower probability of symptomatic infection after intentional exposure to cold viruses [1]. This suggests that social relationship resources may be protective for some infectious diseases. However, human exposure to, and illness from infectious agents, occurs in a complex ecological context that was intentionally excluded from infectious challenge paradigms because all study participants were exposed to the infectious agent [1, 2]. Importantly, this putative salutary link between social relationships and infectious diseases has only been explored with cold viruses and is based upon unplanned analyses from a single study. The potential importance of the association between social networks and communicable diseases warrants further evaluation as it is critical for mitigation.

The present study addresses gaps in our understanding of the links between social resources and infectious diseases by examining the association of social network diversity with colonization by *Staphylococcus aureus* in naturally occurring groups of community residents. *S. aureus* is a bacterium that many humans carry as a commensal, without infections. However, if these bacteria penetrate the outer layers of skin or mucosa, they act as a pathogen and can cause mild skin infections as well as dangerous invasive infections [3]. In 2017 *S. aureus* blood infections caused almost 20,000 deaths in the United States (U.S.) [4]. It is estimated that about a third of the U.S. population is colonized with *S. aureus* [5]. However, our recent work using a combination of culture-based and genomic assays shows a *S. aureus* prevalence of ~66% and suggests that previous prevalence reports are underestimates [6]. While hospital-onset and healthcare-associated invasive infections of methicillin-resistant *S. aureus* (MRSA) have decreased, community-associated MRSA and methicillin-susceptible *S. aureus* (MSSA) infections have not [4, 7]. The epidemiological importance of such a community-based reservoir of *S. aureus* underscores the importance of community sampling to understand prevalence, transmission, and drivers of colonization. Colonization is a major risk factor for infection, with genetic studies showing that colonizing strains of *S. aureus*, i.e., the strains found on skin, nares, etc., are commonly identical to infecting strains, suggesting that patients typically infect themselves [8–10]. Thus, *S. aureus* colonization provides a marker for an infectious disease that is both prevalent and closely tied to clinical infection. Importantly, when assessed in community settings, *S. aureus* colonization remains situated in, and therefore potentially influenced by, the social processes putatively underlying infectious disease exposure.

We examined several social relationship indicators linked to health in general and indicative of social interactions that may drive colonization. These included social network diversity and social network size (the number of people in the social network). Both network diversity and network size are considered measures of social integration, reflecting individual participation in a broad range of social relationships [11] (sometimes labeled social engagement [12]). We complemented these social network metrics with a social integration measure that is used in public health surveillance and only counts interaction with persons outside one's household and disaggregates different forms of social interaction (i.e., phone versus in-person contacts). Importantly, this social integration measure robustly predicts hard health endpoints (i.e., mortality) in large population-based samples and is clearly distinct from socioeconomic status determinants of health [13, 14]. Thus, this social integration measure complements the other social network metrics and provides a bridge to existing population health surveillance.

In addition to these structural social relationship measures this study also assessed loneliness. Loneliness reflects subjective satisfaction with both the quality and quantity of one's

social relationships [15, 16]. Loneliness can therefore disentangle the relative importance of perceived social relationship deficits, which may or may not align with being socially isolated as indicated by quasi-objective social integration scores [15, 17].

Self-rated health was included as a convergent outcome for the analysis of social relationships and *S. aureus* colonization. Self-rated health, typically measured by asking respondents to rate their own health from *poor* to *excellent*, is a robust predictor of mortality [18–21] and captures health status beyond established clinical and biomedical assessments [21, 22]. Because self-rated health is positively associated with several measures of social integration [23, 24], including the public health version we use here [25], self-rated health was used as an additional health status outcome to confirm the convergent validity of the social relationship assessments.

Given the novelty of the outcome variable in relation to social relationship resources, no predictions were made regarding the association of social relationships with *S. aureus* colonization. Instead, several working hypotheses were specified [26]. First, if social network diversity is inversely associated with *S. aureus* colonization, this would extend the literature on salutary social relationships to an infectious agent other than common cold viruses. To the extent that colonization is also inversely associated with other social relationship resources, such as loneliness and/or network size, that would show that the social determinants of infectious disease susceptibility are more extensive than previously understood. If *S. aureus* colonization is unrelated to social network diversity specifically, and to other social relationships generally (i.e., loneliness), this would provide evidence for a boundary for salutary social relationship resources on infectious disease susceptibility. A competing prediction is that social contacts *increase* exposure to persons colonized with *S. aureus*. Therefore the number of contacts, as indicated by social network size, would be positively associated with *S. aureus* prevalence. Similarly, if colonization is unrelated to more frequent social contacts, it does not preclude the possibility that more intimate forms of social interaction (e.g., sexual relationships, cohabitation) may be more likely to result in transmission and colonization. Irrespective of these patterns it was expected that social networks and social integration would be positively associated with self-rated health [23–25].

## Materials and methods

### Study overview

This study is part of a larger investigation of determinants of *S. aureus* colonization and transmission in social groups recruited in south Yuma County in Arizona [6, 27, 28]. Briefly, research team members from the community were trained to recruit participants, to supervise collection of survey data and to supervise participants' self-collected swabs for *S. aureus* culturing. Each participant received a gift card as an incentive. Year 1 was excluded from this analysis as the social network index was only assessed in years 2 and 3 (March 2019 to March 2020). All adult participants provided verbal informed consent to maintain participant anonymity. Consent was witnessed and documented by two staff members. The study (project 1116783) was approved by the Northern Arizona University Institutional Review Board, which also approved verbal rather than written informed consent.

### Participants

Groups of two or more people who appeared to be together were invited by study staff to participate in the study. Study staff confirmed that those approached were part of a self-identified group. If they were not part of a self-identified group they were not invited to participate. No other health screening or eligibility restrictions were required. Children were eligible to

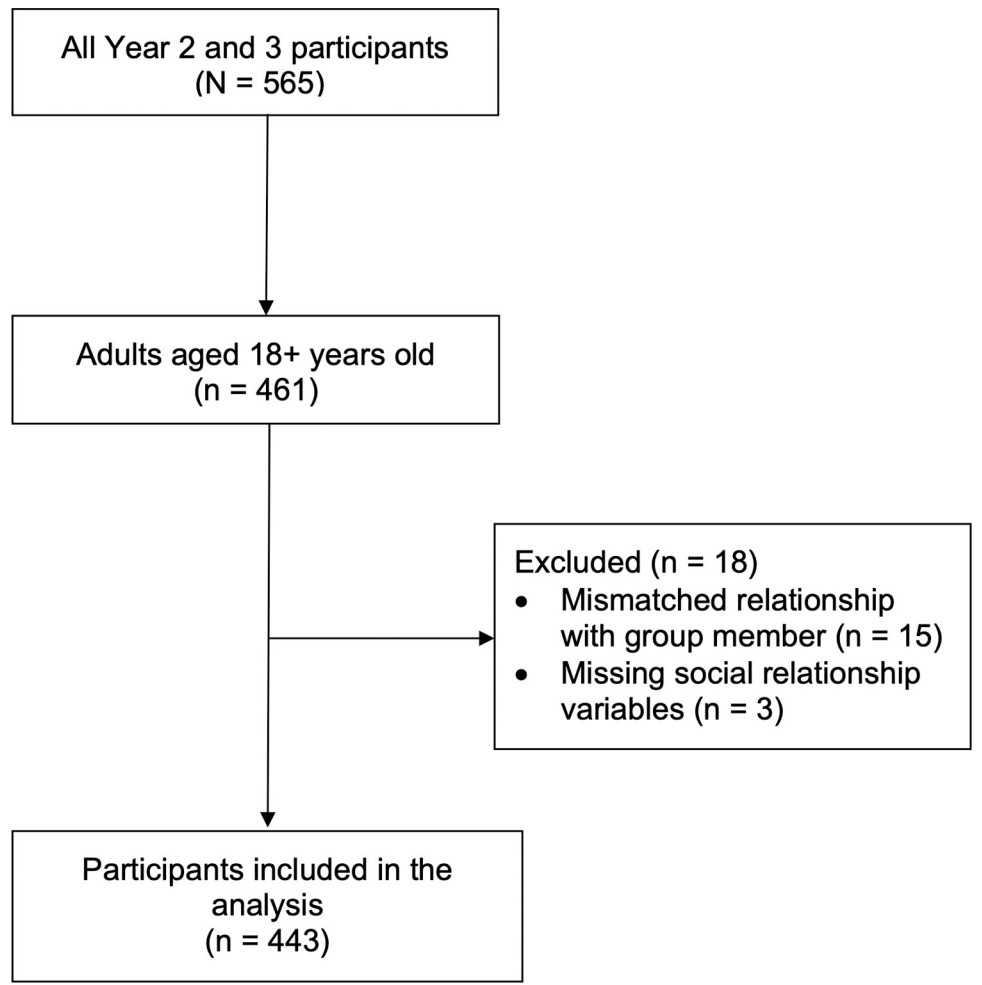

**Fig 1. STROBE flow diagram.**

participate but because their survey responses were often completed by adult proxy respondents, they were excluded and only adults were included. Thus analyses are restricted to adults 18 years and over (Fig 1). The analytic sample size was 443 (442 for self-rated health analyses) and is a subset of a previous study which included all age groups [6]. Race and ethnicity were self-reported by participants using prespecified categories [29].

## Social relationship assessments

**Social network index.** The SNI [1] assesses relationships across 12 types of social roles; spouse/partner, child, in-laws, parents, other close family members, friends, neighbors, coworkers, school contexts, volunteer work, religious contexts, as well as a broad category representing nonreligious social/recreational/professional groups. The SNI also assesses the number of persons interacted with in each social role. If persons interacted (by phone or in person) at least once every two weeks they received a point for that social role domain and these points were summed to create the network diversity score (range 0–12). Network size (the total number of social contacts across those domains) was the sum of all persons with whom the respondent interacted in these roles. Network size counts were naturally limited for some social roles (e.g., 1 person for a spouse/partner role; 2 people each for parents and in-laws) but for other

domains (e.g., close friends) response options ranged from zero to seven or more. The number of persons in the work domain ranged up to 14 as both the number of persons supervised (0–7 +) and the number of nonsupervisory contacts (0–7+) were included in the sum (possible range 0–75). Social network diversity, but not network size, has been linked to infectious disease risk [1].

The last SNI question related to nonreligious group membership was modified from the original [1]. "Fitness classes" was added as an example group activity to make the examples more contemporary and to make the number of contacts assessment parallel to the relatives/ close friends/religious group questions, i.e., top coded at 7 or more persons.

## Social integration and loneliness

Social integration was measured using seven items from the 2001 US National Health Interview Survey (NHIS) [30]. These items capture social contacts with persons not living with the respondent. Questions referenced interactions in the past two weeks and included in-person or phone contact with friends or family, participating in a group social activity, attending a religious service, and eating out. The eating out variable did not refer specifically to eating with others but it aligns statistically with the other social integration items and including it in the index improves stratification of mortality [13]. The last item was whether or not participants were married (also counted in the network diversity score). Questions were scored as yes/no during that interval and yes responses were summed to create a social integration score. This score is strongly associated with all-cause mortality [13, 14] and with psychological well-being [31].

Loneliness was assessed by asking "In general, how often do you feel lonely? Always, usually, sometimes, never." Similar items are used in public health surveys [32] and this single item assessment is strongly associated with depression and depressive symptoms even when statistically controlling for other social relationship variables [33, 34] and loneliness is linked to a weaker immune response to influenza vaccine [35].

## Detection of *S. aureus*

Methods and results for *S. aureus* sampling are detailed in Russakoff et al. [6]. Briefly, two double tipped BBL CultureSwab™ swabs were used for sampling each of three different body sites (nares, throat, and hand). Swabbing was supervised by study staff (who worked in pairs) who also ensured that the self-swabbing was completed properly, i.e., swabbing each site for twenty seconds. Self-swabbing was performed because potential participants are more likely to enroll if they swab themselves and because participants are more likely to swab effectively relative to staff members who might attempt to avoid eliciting a sneeze or causing a gag reflex. Staff supervised swabbing, directing participants to rotate and move the swab around for the full 20 seconds at each site likely further improved sampling effectiveness. The high colonization prevalence is a testament to the success of this method and the quality control protocols. One double tipped swab was used for culturing while the other was used for DNA-based direct detection of *S. aureus*. While in the field, all swabs were stored on ice. In the laboratory, the double tipped swab destined for DNA-based detection was stored at -70˚C before DNA extraction and detection. For direct detection of *S. aureus*, we used SaQuant, a quantitative real-time PCR assay with 95.6% sensitivity and 99.9% specificity [6, 36]. The other double tipped swab, destined for culturing, was stored at 4˚C in the laboratory for no more than 48 hours before culturing to maximize the likelihood of cell survival [37]. For culturing, swabs were streaked onto CHROMagar *S. aureus* media and incubated for 24 hours at 37˚C. CHROMagar Staph aureus chromogenic media allows for differentiation of *S. aureus* from other species as *S.*

*aureus* colonies appear pink to mauve while other bacterial species are inhibited or will appear colorless or blue. Details on efforts to prevent false positive results are detailed in Russakoff et al. [6] and include whole genome sequencing and MLST v2.0 and SCC*mec*Finder v1.2 analyses (www.genomicepidemiology.org) of putative *S. aureus* colonies [38]. For samples that were culture negative but SaQuant positive, an amplicon sequencing approach was used to verify species designation [6] as well as extensive validation of the SaQuant assay to ensure accurate classification on samples with low *S. aureus* quantities [6, 36]. Individuals were recorded as being colonized by *S. aureus* if any of the swabs from any of the three body sites tested positive for *S. aureus* using the SaQuant assay or culture. Culture positive samples where the paired double tipped swab tested negative for SaQuant were counted as negative if the culture was confirmed as not *S. aureus* by whole genome sequencing.

## Data reduction and analysis

To ensure reasonable numbers of positive *S. aureus* cases in social relationship strata, social relationship variables were collapsed into smaller categories. A 6 category social integration variable was created by combining the lowest 4 groups (i.e., 0–3, 4, 5, 6, 7 & 8), a 6 category network diversity score was created by collapsing the lowest and highest categories (0–4, 5, 6, 7, 8, 9–12) and the *often* and *always* lonely groups were also combined. Similarly, the network size variable was reduced to 5 categories with approximately equal numbers in each category (cutoffs of 0–15, 16–21, 22–25, 26–33, 34–65) (Table 1). All social relationship variables met an interval assumption (could be modeled as a single variable with multiple levels) [39] and were therefore entered as individual terms in all regression models.

Each social relationship variable was examined individually to predict binary *S. aureus* colonization, adjusting for age and sex. The fully adjusted models were of primary interest but unadjusted models are also reported. The main model was a generalized linear model using a Poisson family and a log link with robust clustered variance estimates. This model estimates prevalence ratios (labeled here incidence rate ratios) which are preferable to odds ratios derived from logistic regression [40], particularly when the outcome is common [41, 42]. Odds ratios also have the disadvantage of incorporating an arbitrary scaling factor that precludes odds ratio comparisons across models with different explanatory variables within a study, or comparing odds ratios across different studies [41, 43]. The clustered variance option adjusts the statistical tests to accommodate the group-level sampling. A two sided p-value of 0.05 was used to determine statistical significance. Stata 16.1 (Stata Corp) was used for all analyses. Post-hoc power analysis for 191 clusters (social groups) with a median cluster size of N = 3 and an intraclass correlation of *S. aureus* colonization of 0.14 has power > 0.89 to detect a 15% colonization difference.

## Results

Sociodemographic characteristics of participants are in Table 1. Most participants (60.7%) were married, a slight majority (52.6%) were female, most (68.4%) reported Hispanic ethnicity and about a third (30%) chose to complete the Spanish version of the survey. Correlations among the social relationship indicators, self-rated health (SRH) and colonization with *S. aureus* are in Table 2.

### Social relationships and self-rated health

Ordinary least squares regression models with clustered variance estimates were used to predict SRH with the 4 social relationship measures. Each of these was associated with SRH in adjusted models and all but social integration were associated in unadjusted models (Table 3).

**Table 1. Sample sociodemographic characteristics.**

| Characteristic | Total | *S. aureus* negative | *S. aureus* positive |
|---|---|---|---|
| Participants, n | 443 | 156 | 287 |
| Age y, mean (SD) | 36.4 (15.1) | 38.7 (16.7) | 35.2 (14.0) |
| Female sex, % (N) | 52.6 (233) | 66.0 (103) | 45.3 (130) |
| Hispanic ethnicity, % (N) | | | |
| No | 29.8 (132) | 20.5 (32) | 34.8 (100) |
| Yes | 68.4 (303) | 75.6 (118) | 64.5 (185) |
| Missing | 1.8 (8) | 3.8 (6) | 0.7 (2) |
| Race, % (N) | | | |
| White | 66.8 (296) | 62.8 (98) | 69.0 (198) |
| Black | 1.1 (5) | 1.3 (2) | 1.0 (3) |
| Native American | 0.5 (2) | 0.6 (1) | 0.3 (1) |
| Asian | 1.6 (7) | 1.9 (3) | 1.4 (4) |
| Pacific Islander | 0.2 (1) | 0.0 (0) | 0.3 (1) |
| Other race | 4.3 (19) | 7.1 (11) | 2.8 (8) |
| Multiple race | 3.8 (17) | 2.6 (4) | 4.5 (13) |
| No preferred race | 9.9 (44) | 9.6 (15) | 10.1 (29) |
| Missing | 11.7 (52) | 14.1 (22) | 10.5 (30) |
| Education level, % (N) | | | |
| < High school | 12.0 (53) | 17.9 (28) | 8.7 (25) |
| High school diploma | 14.0 (62) | 16.0 (25) | 12.9 (37) |
| Some college | 50.8 (225) | 40.4 (63) | 56.4 (162) |
| College graduate or higher | 23.0 (102) | 25.0 (39) | 22.0 (63) |
| Missing | 0.2 (1) | 0.6 (1) | 0.0 (0) |
| Marital status, % (N) | | | |
| Married/cohabiting | 60.7 (269) | 51.3 (80) | 65.9 (189) |
| Divorced/separated | 5.9 (26) | 7.7 (12) | 4.9 (14) |
| Widowed | 1.4 (6) | 3.2 (5) | 0.3 (1) |
| Never married | 25.3 (112) | 26.3 (41) | 24.7 (71) |
| Missing | 0.9 (4) | 1.3 (2) | 0.7 (2) |
| Employed, % (N) | 78.8 (349) | 71.2 (111) | 82.9 (238) |
| Language use at home, % (N) | | | |
| Only Spanish | 17.2 (76) | 27.6 (43) | 11.5 (33) |
| More Spanish than English | 13.3 (59) | 15.4 (24) | 12.2 (35) |
| Equal Spanish and English | 23.3 (103) | 21.8 (34) | 24.0 (69) |
| More English than Spanish | 13.8 (61) | 14.1 (22) | 13.6 (39) |
| Only English | 32.3 (143) | 21.2 (33) | 38.3 (110) |
| Other | 0.2 (1) | 0.0 (0) | 0.3 (1) |
| Language of survey, % (N) | | | |
| Spanish | 30.0 (133) | 44.2 (69) | 22.3 (64) |
| English | 70.0 (310) | 55.8 (87) | 77.7 (223) |
| Social integration score, % (N) | | | |
| 1–3 | 6.3 (28) | 9.0 (14) | 4.9 (14) |
| 4 | 6.5 (29) | 5.8 (9) | 7.0 (20) |
| 5 | 20.1 (89) | 21.2 (33) | 19.5 (56) |
| 6 | 25.3 (112) | 22.4 (35) | 26.8 (77) |
| 7 | 28.2 (125) | 30.1 (47) | 27.2 (78) |
| 8 | 13.5 (60) | 11.5 (18) | 14.6 (42) |
| Social network diversity score, % (N) | | | |
| 0–4 | 18.3 (81) | 19.9 (31) | 17.4 (50) |

*(Continued)*

**Table 1.** (Continued)

| Characteristic | Total | *S. aureus* negative | *S. aureus* positive |
|---|---|---|---|
| 5 | 17.8 (79) | 17.3 (27) | 18.1 (52) |
| 6 | 16.9 (75) | 18.6 (29) | 16.0 (46) |
| 7 | 17.4 (77) | 19.9 (31) | 16.0 (46) |
| 8 | 14.0 (62) | 12.2 (19) | 15.0 (43) |
| 9–12 | 15.6 (69) | 12.2 (19) | 17.4 (50) |
| Social network size, % (N) | | | |
| 0–15 | 15.1 (67) | 17.9 (28) | 13.6 (39) |
| 16–21 | 20.1 (89) | 21.8 (34) | 19.2 (55) |
| 22–25 | 16.9 (75) | 12.8 (20) | 19.2 (55) |
| 26–33 | 24.6 (109) | 27.6 (43) | 23.0 (66) |
| 34–65 | 23.3 (103) | 19.9 (31) | 25.1 (72) |
| Loneliness, % (N) | | | |
| Never | 57.1 (253) | 60.3 (94) | 55.4 (159) |
| Sometimes | 36.3 (161) | 31.4 (49) | 39.0 (112) |
| Often/always | 6.5 (29) | 8.3 (13) | 5.6 (16) |
| Self-rated health, % (N) | | | |
| Poor/fair | 11.1 (49) | 16.0 (25) | 8.4 (24) |
| Good | 36.6 (162) | 35.3 (55) | 37.3 (107) |
| Very good | 36.6 (162) | 32.1 (50) | 39.0 (112) |
| Excellent | 15.6 (69) | 16.0 (25) | 15.3 (44) |
| Missing | 0.2 (1) | 0.6 (1) | 0.0 (0) |

Social integration score was derived from 8 items from the US National Health Interview Survey. Social network diversity and network size were measured using the Social Network Index [1].

Overall, the social relationship resources were moderately correlated with one another, and as in other literature [23–25], were predictive of higher self-rated health. Similar findings were observed in sensitivity analyses using an ordinal logit model. Raw data are provided in S1 File.

## Colonization with *S. aureus*

Overall 64.8% (*n* = 287) of the participants were colonized with *S. aureus*. Sample characteristics by *S. aureus* status are in Table 1. Raw data are provided in S1 File.

## Social relationships and colonization with *S. aureus*

Neither social network diversity nor social network size were associated with *S. aureus* colonization (Table 4). The NHIS social integration variable and loneliness were also unrelated to *S. aureus* colonization. These patterns were consistent with and without adjustment for age and sex (Table 4). Log binomial models [44] predicting *S. aureus* colonization were almost identical to Poisson estimates and were also consistent when using indicator social relationship variables. Similar patterns were seen when using continuous social relationship variables (Table 4) and when covarying socioeconomic status markers (education and home ownership).

## Discussion

This study examined the association between *S. aureus* colonization and a number of social relationship resources that may denote exposure to infectious agents. There was no association

**Table 2. Correlations among social relationship variables, self-rated health and colonization with *S. aureus*.**

|  | Network size | Social integration | Lonely | Self-rated health | *S. aureus* colonized |
|---|---|---|---|---|---|
| Social network diversity | 0.69 | 0.53 | -0.11 | 0.17 | 0.06 |
|  | (<0.001) | (<0.001) | (0.019) | (0.001) | (0.198) |
| Social network size |  | 0.36 | -0.11 | 0.19 | 0.06 |
|  |  | (<0.001) | (0.045) | (<0.001) | (0.226) |
| Social integration |  |  | -0.19 | 0.09 | 0.05 |
|  |  |  | (<0.001) | (0.073) | (0.254) |
| Lonely |  |  |  | -0.19 | 0.02 |
|  |  |  |  | (<0.001) | (0.730) |
| Self-rated health |  |  |  |  | 0.07 |
|  |  |  |  |  | (0.159) |

*P*-values in parentheses. Estimates adjusted for clustering. N's = 442–443

of social network diversity with *S. aureus* colonization prevalence within naturally occurring social groups recruited from the community. *S. aureus* prevalence was also unrelated to loneliness, social network size, and social integration. These social relationship variables have previously been linked to a number of health outcomes including all-cause mortality [13, 23, 45]. In contrast, convergent validity was observed for all social relationship indicators via their positive associations with global SRH, the latter of which is both a clinical endpoint and robustly associated with mortality [20, 21, 46].

The hypothesis that more social interaction, and thus greater exposure, would be associated with higher *S. aureus* prevalence, was not supported. This was a strong test of the hypothesis given the sample reported a large average social network size and given that the association of network size and *S. aureus* colonization was nonsignificant and of very small magnitude. Household-level contacts are commonly linked to colonization and transmission of *S. aureus* [47–49], however little is known about the impact of social relationships outside the home or healthcare settings [50]. These results support the contention that social interactions outside the home (typical of social network assessments) are less important than those within the home for *S. aureus* colonization.

These data show that the salutary association observed between social network diversity and susceptibility to cold infection (found in only one study to date) does not extend to colonization with *S. aureus*. In prior work with colds, all participants were intentionally exposed to one of two cold viruses [1] whereas in the present study, exposure was naturally occurring. Thus, infectious challenge studies do not directly evaluate the link between social ties and a) the likelihood of coming in contact with an infected person or b) the chance of becoming

**Table 3. Coefficients regressing self-rated health on social relationship variables.**

|  | Unadjusted | | | Adjusted for age and sex | | |
|---|---|---|---|---|---|---|
|  | b | (95%CI) | p | b | (95%CI) | p |
| Social network diversity | 0.09 | (0.04–0.14) | 0.001 | 0.08 | (0.03–0.13) | 0.001 |
| Social network size | 0.12 | (0.06–0.19) | <0.001 | 0.10 | (0.04–0.16) | 0.001 |
| Social integration | 0.06 | (-0.01–0.13) | 0.073 | 0.07 | (0.00–0.13) | 0.044 |
| Lonely | -0.28 | (-0.41– -0.14) | <0.001 | -0.32 | (-0.45– -0.19) | <0.001 |

Ordinary least squares regression adjusted for clustering. N = 442

CI = confidence interval

**Table 4. Incidence rate ratios (IRR) regressing *S. aureus* colonization on individual social relationship variables.**

| | Unadjusted | | | Adjusted for age and sex | | |
|---|---|---|---|---|---|---|
| | **IRR** | **(95%CI)** | **p** | **IRR** | **(95%CI)** | **p** |
| **Categorical variables** | | | | | | |
| Social network diversity | 1.03 | (0.99–1.07) | 0.20 | 1.02 | (0.98–1.06) | 0.26 |
| Social network size | 1.03 | (0.98–1.09) | 0.23 | 1.02 | (0.97–1.07) | 0.56 |
| Social integration | 1.03 | (0.98–1.08) | 0.26 | 1.03 | (0.98–1.08) | 0.24 |
| Lonely | 1.02 | (0.91–1.14) | 0.73 | 1.02 | (0.91–1.14) | 0.74 |
| **Continuous variables** | | | | | | |
| Social network diversity | 1.03 | (0.99–1.06) | 0.12 | 1.02 | (0.99–1.06) | 0.15 |
| Social network size | 1.00 | (1.00–1.01) | 0.26 | 1.00 | (1.00–1.01) | 0.59 |
| Social integration | 1.03 | (0.98–1.07) | 0.29 | 1.03 | (0.98–1.08) | 0.26 |

All estimates are adjusted for clustering. N = 443

CI = confidence interval

infected given that contact [2, 51]. The present study did not examine clinical infection but preserved these two critical steps on the pathway, with step 2 being colonization rather than infection. Importantly, *S. aureus* colonization is outside the purview of immune system surveillance and thus is not informed by psychoneuroimmunology models.

This study is unique in that it examined the above mentioned social relationship resources in conjunction with *S. aureus* colonization in people living in a diverse border setting in southern Arizona. Potential participants were recruited from, and completed participation in, public and private spaces. Enrollees completed social relationship assessments and provided swabs from three body sites (hand, anterior nares, and throat) for *S. aureus* detection. No other health screening or eligibility restrictions were required. Thus, this data collection closely parallels the ecology from which *S. aureus* infection commonly arises and adds to our understanding of community transmission and prevalence by showing that social contacts as commonly assessed do not drive colonization. In a larger context, this work extends prior work on the link between social network resources and infection a) by examining a different infectious agent; b) by incorporating a broad assessment of social relationships; and c) by sampling within a multiethnic bilingual population.

This study has a number of strengths and limitations to consider. Colonization prevalence was substantially higher in this sample [6] relative to other estimates [5]. This is likely because of the greater sensitivity due to sampling from 3 different body sites and coupling molecular and culture-based detection methods, approaches that in combination are unique to this study [6, 52]. Greater colonization prevalence also indicates the effectiveness of self-swabbing quality controls. This study used the same measure of network diversity as in prior studies of infectious disease risk, and complemented that measure with additional social integration assessments and a subjective social relationship measure. As in other studies [53] correlations among these measures of quasi-objective and subjective social relationship resources were of small to medium magnitude (except for network size and diversity measures, which were

strongly correlated with each other) and thus provide good coverage of different social relationship domains. The distribution of social integration in the present sample was similar to representative samples [14]. We had to collapse the lowest social integration categories but this was expected as very low social integration is rare, observed in less than 2% of the population [13].

This study was cross-sectional and did not include a clinical infectious outcome. It is therefore possible that clinical *S. aureus* infection, rather than colonization, is linked to social relationships. Recent work indicates that loneliness, but not social isolation, has a small but statistically significant association with a broad index of hospital-treated infections [54]. However, in that study loneliness was not associated with infections often caused by *S. aureus*, i.e., skin and soft tissue infections [55], thus providing convergent evidence that neither colonization nor putative *S. aureus* clinical infection are related to these social relationship resources. On a community level, this investigation of social relationship determinants of carriage in naturally occurring social groups, using a rigorous assessment of *S. aureus* colonization, illuminates the extent to which social relationships stratify community reservoirs of this important infectious agent.

## Conclusions

In a sample of naturally occurring social groups within a community, this study did not find a statistically significant association between social network diversity and *S. aureus* colonization prevalence. The results also did not support the hypothesis that more social interaction would be associated with a higher *S. aureus* prevalence. Coupled, these findings suggest that, in this sample, structural social relationship characteristics and loneliness are not associated with colonization of *S. aureus*. Future research on *S. aureus* colonization should examine more intimate forms of social contact and longitudinal carriage to determine sources of transmission and ultimately, targeted mitigation strategies for persons at risk of infection.

## Supporting information

**S1 File. Raw data for the study.**
(XLS)

## Acknowledgments

This work would not have been possible without the support from the Regional Center for Border Health in Somerton, AZ. The authors are grateful to the many field surveyors from NAU and the Regional Center for Border Health who recruited, sampled, and collected data from participants.

## Author Contributions

**Conceptualization:** Steven D. Barger, Monica R. Lininger, Robert T. Trotter, II, Talima Pearson.

**Data curation:** Steven D. Barger, Monica R. Lininger, Mimi Mbegbu, Shari Kyman, Kara Tucker-Morgan, Colin Wood, Briana Coyne, Sara Maltinsky, Talima Pearson.

**Formal analysis:** Steven D. Barger, Monica R. Lininger, Robert T. Trotter, II.

**Funding acquisition:** Steven D. Barger, Monica R. Lininger, Robert T. Trotter, II, Talima Pearson.

**Investigation:** Steven D. Barger, Monica R. Lininger, Mimi Mbegbu, Shari Kyman, Kara Tucker-Morgan, Colin Wood, Briana Coyne, Benjamin Russakoff, Kathya Ceniceros, Cristina Padilla, Sara Maltinsky, Talima Pearson.

**Methodology:** Steven D. Barger, Monica R. Lininger, Robert T. Trotter, II, Mimi Mbegbu, Shari Kyman, Colin Wood, Briana Coyne, Talima Pearson.

**Project administration:** Robert T. Trotter, II, Mimi Mbegbu, Shari Kyman, Kara Tucker-Morgan, Sara Maltinsky, Talima Pearson.

**Supervision:** Mimi Mbegbu, Shari Kyman, Kara Tucker-Morgan, Sara Maltinsky, Talima Pearson.

**Validation:** Steven D. Barger, Monica R. Lininger, Robert T. Trotter, II, Talima Pearson.

**Writing – original draft:** Steven D. Barger, Monica R. Lininger, Talima Pearson.

**Writing – review & editing:** Robert T. Trotter, II, Mimi Mbegbu, Shari Kyman, Kara Tucker-Morgan, Colin Wood, Briana Coyne, Benjamin Russakoff, Kathya Ceniceros, Cristina Padilla, Sara Maltinsky.

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
