## [Decision Letter · Decision Letter 0]

31 Jan 2023

PONE-D-23-00421Cross-sectional study of the association of social relationship resources with Staphylococcus aureus colonization in naturally occurring social groupsPLOS ONE

Dear Dr. Steven D Barger, 

Thank you for submitting your manuscript to PLOS ONE. After careful consideration, we feel that it has merit but does not fully meet PLOS ONE’s publication criteria as it currently stands. Therefore, we invite you to submit a revised version of the manuscript that addresses the points raised during the review process.

ACADEMIC EDITOR: 

Dear author,

Please read the comments of both reviewers carefully, revise the manuscript in light of reviewer's comments. Submit the revised manuscript and rebuttal letter. The submitted revised manuscript will be re-reviewed before acceptance.

We look forward to receiving your revised manuscript.

Kind regards,

Samiullah Khan, Ph. D

Academic Editor

PLOS ONE

Reviewers' comments:

Reviewer's Responses to Questions

**Comments to the Author**

1. Is the manuscript technically sound, and do the data support the conclusions?

Reviewer #1: Partly

Reviewer #2: Partly

2. Has the statistical analysis been performed appropriately and rigorously? 

Reviewer #1: No

Reviewer #2: No

3. Have the authors made all data underlying the findings in their manuscript fully available?

Reviewer #1: Yes

Reviewer #2: Yes

4. Is the manuscript presented in an intelligible fashion and written in standard English?

Reviewer #1: Yes

Reviewer #2: No

5. Review Comments to the Author

Reviewer #1: This relates to a manuscript entitled Cross-sectional study of the association of social relationship resources with Staphylococcus aureus colonization in naturally occurring social groups.

General comment

Topic of the manuscript is attractive to certain extent. But has some grey areas that need improvisation.

It is well known that mere colonization of S. aureus will not lead to any kind of infections in a health population. It is mainly associated with patient and microbial related factors. eg. Hospitalization/Immunodeficiency/metabolic disorders/ drug resistance- MRSA/VRSA

Since the study population is healthy, what is the rationale of this research

Moreover, in the introduction the authors have mentioned that i/3rd of the US population is colonized with S. aureus.

Mode of transmission and factors that trigger the transmission rate associated with S. aureus is already know. And S. aureus infection is not a contagious disease too. By maintaining proper hand hygiene or by decolonization this can be managed to a greater extent.

Above all, the results of the research are inconclusive and shallow.

In this context, what is the relevance of this research?

Specific comment

Strictly follow the journal’s instructions for formatting the abstract and other parts

The English portion of the article need to be improved

Title

Modify the title by precisely showing the study settings and study population

Abstract

Has to be narrate properly. Methodology and conclusion part needs to be re-written

Introduction

Introduction: It needs to be more informative

It is too general, broad and shallow.

Organize according to the objectives of the study.

Update the reference

Rationale has to be more emphasized.

State the precise problems existing in the study population or area

Research gap existing in the study area.

Materials and methods

1. It is mandatory to include the details sample size calculation

2. Inclusion and exclusion criteria has to be precisely mentioned.. Why authors excluded children

3. Have you included any participants under decolonization procedure

4. Study was conducted during the wake of covid outbreaks… Have you taken any precautionary measures for collecting nare samples Ɂ

5. Why you prefer self-sampling

6. How you maintain the quality of sample

7. Statistical analysis is unclear

8. Cite the Tables properly

9. Have you checked antimicrobial susceptibility profile of positives cases with regard to MRSA/VRSA

10. Have authors recommended decolonization of positive cases

11. Status of MRSA and VRSA and co-resistance profiles have to be included

Results

Please pinpoint the most relevant results rather than incurious findings

Discussion

Need substantial improvement.

Please improve the main conclusion of the manuscript.

Reviewer #2: 1. The study appears to be well-conducted, but I recommend updating the references section to include more recent research on the topic. This will provide a more complete and current understanding of the subject and will also help to strengthen the overall credibility of the study.

2. The results of the manuscript appear to be weak and it would be beneficial to update the study with more recent findings. It is also important to highlight the use of whole genome sequencing in the results section, as it is a significant aspect of the study that should be emphasized and discussed in relation to the findings. This will help to strengthen the overall validity and impact of the study.

6. PLOS authors have the option to publish the peer review history of their article (what does this mean?). If published, this will include your full peer review and any attached files.

Reviewer #1: No

Reviewer #2: No

---

## [Author Response · Author response to Decision Letter 0]

4 Mar 2023

March 4, 2023

Samiullah Khan, Ph.D 

Academic Editor 

PLOS ONE 

Dear Dr. Khan:

We are grateful to have the opportunity to revise our PLOS ONE submission titled “Cross-sectional study of the association of social relationship resources with Staphylococcus aureus colonization in naturally occurring social groups” and we are thankful to the Editor and Reviewers for their feedback. Our responses to Editor requests and reviewers’ comments are below. As instructed, we uploaded three files comprising the rebuttal letter, the revised document with tracked changes, and the revised document without tracked changes. We also included a supplemental file containing the raw data used for our study.

Editor comments:

We reviewed the PLOS ONE style requirements and believe we have complied with those guidelines.

2. Please provide additional details regarding participant consent. In the ethics statement in the Methods and online submission information, please ensure that you have specified what type you obtained (for instance, written or verbal, and if verbal, how it was documented and witnessed). If your study included minors, state whether you obtained consent from parents or guardians. 

We note that participants provided verbal informed consent that was documented by two staff members. We now include the IRB project number in the Study Overview section. This manuscript was restricted to persons 18 years of age or older.

3. We note that you have stated that you will provide repository information for your data at acceptance. Should your manuscript be accepted for publication, we will hold it until you provide the relevant accession numbers or DOIs necessary to access your data. 

We included the raw data as a supplementary file with the resubmission. It is titled S1 Dataset. 

4. We note that you have included the phrase “data not shown” in your manuscript. Unfortunately, this does not meet our data sharing requirements. PLOS does not permit references to inaccessible data. … Or, if the data are not a core part of the research being presented in your study, we ask that you remove the phrase that refers to these data.

We did not intend to refer to inaccessible data, but rather to sensitivity analyses of the main data using different statistical models. We apologize that this was unclear. Those analyses confirmed the findings from the main statistical models, but were not included in the manuscript. We report these sensitivity analyses to increase the rigor of our study, i.e., to demonstrate our findings were not driven by the analytic approach. We removed the phrasing about not showing the data and also include the raw data in the resubmission. 

In the brief checklist the reviewers disagreed with the statement that the statistical analysis was “performed appropriately and rigorously.” Although no specific examples were provided, we highlight our optimal statistical practices here. First, for the binary colonization outcome variable we used a generalized linear model to generate rate ratios. This is preferable to a logistic regression model producing odds ratios, as odds ratios are biased when the base rate of the binary outcome is > 10% (JAMA, 2018. 320(1): p. 84-8; Am J Epidemiol, 2004. 159(7): p. 702-6) and do not produce coefficients that are comparable across statistical models, either within or across studies. As shown in the Zou (2004) reference, our GLM approach is unbiased and has good coverage of the nominal 95% confidence intervals. We elaborated on the disadvantages of odds ratios in the revised manuscript, adding additional references to support this approach. In fact, two members of our team (SDB and MRL) recently published a primer describing optimal analytic practices for binary outcomes-that reference is included in the revision.

There are additional analytic features that indicate rigor. We incorporated clustering in our analyses because the potential lack of independence of our participants, who were recruited in small groups. We evaluated the functional form of the explanatory variables in the models, confirming appropriate model specification. We also tested alternative but related statistical models for both colonization and self-rated health (log binomial and cumulative logit models, respectively) which confirmed the main analyses. We are confident these data are analyzed transparently, optimally, and that our findings are not driven by arbitrary analytic or data reduction choices. 

Reviewer comments

Reviewer #1: This relates to a manuscript entitled Cross-sectional study of the association of social relationship resources with Staphylococcus aureus colonization in naturally occurring social groups.

General comment

Topic of the manuscript is attractive to certain extent. But has some grey areas that need improvisation.

It is well known that mere colonization of S. aureus will not lead to any kind of infections in a health [sic] population. It is mainly associated with patient and microbial related factors. eg. Hospitalization/Immunodeficiency/metabolic disorders/ drug resistance- MRSA/VRSA

Since the study population is healthy, what is the rationale of this research

Moreover, in the introduction the authors have mentioned that i/3rd of the US population is colonized with S. aureus.

1. Unfortunately, this reviewer’s understanding of the problems, challenges, and knowledge gaps is not aligned with the consensus opinions of public health agencies and researchers. While colonization of S. aureus does not necessarily lead to infection, it is a major risk factor. This principal has guided mitigation practices that have reduced infections in health care populations over the last 15 years. Unfortunately, progress in healthcare settings has slowed, and an increase in community acquired cases is cause for alarm and suggestive of an important reservoir outside of the healthcare setting and into the community (Kourtis et al. MMWR. 2019;68:214-9.). For this reason, research is now increasingly being focused on understanding infections and the spread of S. aureus in the community. Our previous work has established that ~65% of community members in southern Arizona may be colonized, with significant ethnic and sex-based disparities. This prevalence is much higher than previously reported, and likely reflects a combination of singleton site sampling and the inadequacy of culture-based colonization ascertainment in prior studies. In fact, our earlier work shows that almost half of culture-negative assays were positive for S. aureus when evaluated with qPCR (Russakoff et al. J Infect Dis 2022). 

We have a very poor understanding about who is colonized, why they may be colonized, and whether the pathogen is likely to spread to other body sites, be transmitted to other people, or cause infections. Our work here is to address perhaps the most important hypothesis about the spread of an infectious disease – whether social interactions are likely to result in spread and higher rates of colonization. Our results are definitive in that the types of social interactions studied here do not impact the likelihood of carriage. This has an important bearing on design of future studies and mitigation efforts. These points are all made in the introduction and in the abstract.

Mode of transmission and factors that trigger the transmission rate associated with S. aureus is already know [sic]. 

2. This is definitely not true. We have very little understanding of mechanisms and precise sources of transmission and even transmission rates. Our qPCR validation assays show that we do not accurately understand even carriage in the population, let alone transmission modes (see above). This is why community-acquired infections are on the rise and likely why even the effectiveness of healthcare-associated infection control efforts have plateaued. Even sampled body sites are inconsistent from study to study, thus obscuring the true prevalence in the community. We have two funded NIH grants (see Funding section) to address these significant needs and other funding agencies are pouring significant amounts of money into research to address these very questions which are foundational to efforts that explore decolonization and pathogen reduction as a strategy to prevent infection (see Topics 22.4 and 22.5 in the latest CDC BAA which can be downloaded at this site https://sam.gov/opp/15229982f7c348f69fd35e9a0add8aba/view). (Also see the FDA and CDC sponsored workshop: https://info.rescueagency.com/en-us/drug-development-consideration-virtual-public-workshop-cdc-fda#description).

And S. aureus infection is not a contagious disease too. 

3. This is not true, although we are a little confused by this statement as it is the bacteria that are contagious and not the disease. While we know that S. aureus can be transmitted, we have only a very rudimentary understanding of the mechanisms. Importantly, this work is about carriage and not infection. We make this distinction throughout the paper (including in the very first sentence of the abstract) and discuss why understanding carriage has important implications on infections.

By maintaining proper hand hygiene or by decolonization this can be managed to a greater extent.

4. We are surprised that the reviewer makes this point about controlling contagious diseases after incorrectly stating in his/her previous point that S. aureus is not contagious. Unfortunately, the infection rate data (Kourtis et al. 2019) and the request by funding agencies for research to better understand and mitigate the spread of S. aureus demonstrate that this statement is inaccurate. There is very little evidence on what body sites act as sources, but current understanding indicates the hands are probably not a significant source. Decolonization has a lot of potential, but has thus far fallen short of expectations because we need to better understand community carriage. The necessity of this point is evidenced by the latest call for proposals by the CDC soliciting work to better characterize carriage so that decolonization can be more efficient. Please see Topic 22.5 in the CDC BAA which can be downloaded at this site (https://sam.gov/opp/15229982f7c348f69fd35e9a0add8aba/view). More broadly, community onset infection is an important element of ongoing S. aureus surveillance efforts in the U.S., which encompasses 16 million persons (https://www.cdc.gov/hai/eip/saureus.html).

Above all, the results of the research are inconclusive and shallow.

5. We respectfully disagree. Our research is conclusive that the types of social interactions that we consider are not correlated with the likelihood of being colonized. We’re not sure what the reviewer means by “shallow”. As shown in other studies, our social relationship assessments have substantial prognostic value for clinical health endpoints including depression, infectious disease susceptibility, and mortality. Thus, our battery of social relationship assessments is optimal, and provides a strong test of whether these are associated with S. aureus colonization. As noted below, any association of e.g., loneliness with clinical infection is probably quite small. 

In this context, what is the relevance of this research?

5. Please refer to our point #1 above. 

Specific comment

Strictly follow the journal’s instructions for formatting the abstract and other parts 

6. As noted above we reviewed the manuscript and it should conform to formatting guidelines.

The English portion of the article need [sic] to be improved 

7. All co-authors reviewed the document and provided editorial input. We feel that the article is written clearly and succinctly. Given that this reviewer did not provide specific examples we did not undertake any English-language revision. 

Title

Modify the title by precisely showing the study settings and study population 

8. The title describes the study design, the research question and the small group sampling approach. We added “along the US/Mexico border” to the title to further identify the setting. 

Abstract

Has to be narrate [sic] properly. Methodology and conclusion part needs to be re-written 

9. Without specific examples, this comment is not actionable. We have clearly defined subsections of the methodology to organize the manuscript in a reader-friendly manner. 

Introduction

Introduction: It needs to be more informative 

It is too general, broad and shallow. 

10. These are sweeping statements that are not actionable without specifying what elements the reviewer feels need more coverage. We believe that we have provided sufficient background to: a) understand the competing ways in which social network resources can impact health, b) understand the relative aspects of S. aureus biology and epidemiology, c) describe and justify the social network assessments that we used, and d), describe the potential implications of the work depending on different possible outcomes. We clearly delineate multiple competing hypotheses in the introduction, which aligns with the approach found in other studies of social relationships and infectious pathogens (Lancet Public Health. 2023;8: e109-e18).

Organize according to the objectives of the study. 

11. There is only one objective to this study which is laid out clearly in the title and abstract. We use the introduction to provide background for this objective.

Update the reference 

12. Our reference #35 referred to a submitted manuscript. This was an error as it was published in 2021. We corrected this. More broadly, our current understanding of S. aureus carriage in the US population is largely based on the 2001–2002 National Health and Nutrition Examination Survey (NHANES), hence that reference is from 2008. Unfortunately, there aren’t more recent population-based estimates. We do however contrast those findings with our own recent work that we discuss and cite. Our revised manuscript also incorporates a paper published after we submitted our manuscript. That paper evaluates the association of social isolation and loneliness with infectious disease related hospitalization (see last paragraph of Discussion section). Requests to “update” other references are not applicable as these citations are for established theoretical models of social relationships (Brissette; Berkman).

Rationale has to be more emphasized. 

13. We believe that the rationale is clearly laid out in the abstract and introduction. Furthermore, at the end of the introduction, we discuss how different outcomes will have different implications on our understanding of S. aureus transmission and will translate into more efficient and targeted mitigation practices (see also point 1 above). By describing and testing several working hypotheses (last paragraph of Introduction), we model optimal scientific practices (Science 1964;146(3642):347-53), avoiding the confirmatory biases that underlie most scientific literature (Scientometrics (2012) 90:891–904; Nat Hum Behav (2019) 3:197). In fact, we chose PLOS ONE as a publication outlet because its policies are specifically designed to remedy systemic biases in scientific publication. 

State the precise problems existing in the study population or area 

14. We are confused by this comment as this work is not about identifying problems in a specific study population, but rather, using a population to test a specific hypothesis that is well laid out in the introduction and abstract. 

Research gap existing in the study area. 

15. The knowledge gaps that this work addresses are well described in the introduction and abstract. For example, in the 2nd paragraph of the introduction we note “…we address gaps in our understanding of the links between social resources and infectious diseases by examining the association of social network diversity with colonization by Staphylococcus aureus in naturally occurring groups of community residents.” It is unknown the extent to which social connections are linked to infectious pathogens in naturally occurring social groups.

Materials and methods

It is mandatory to include the details sample size calculation

16. We did not conduct a priori power calculations as we had no prior estimate of the group-level intraclass correlation for staph colonization. Our data collection was also prematurely terminated due to the COVID lockdown. However, in the revised manuscript we now include post hoc power calculations, showing substantial power to detect colonization differences on par with the ethnic disparity seen in our data. 

Inclusion and exclusion criteria has to be precisely mentioned.. Why authors excluded children

17. In the participant section, we state that “groups of two or more people who appeared to be together were invited by study staff to participate in the study”. We also state the reason why children were excluded (“Children were eligible to participate but because their survey responses were often completed by adult proxy respondents we restrict our analysis to adults 18 years and over”). These were the only inclusion/exclusion criteria. We also provide a citation to another report in case a reader would like details that may not be pertinent here.

Have you included any participants under decolonization procedure

18. None of the participants told us that they were undergoing decolonization for S. aureus. 

Study was conducted during the wake of covid outbreaks… Have you taken any precautionary measures for collecting nare samples Ɂ 

19. We did not as all samples were collected before March, 2020 (pre-COVID). Study dates have been added to the Study Overview section. 

Why you prefer self-sampling 

20. We have added a statement about this in the text. In short, participants are more likely to participate if they can control sampling. Most importantly, self-sampling was clearly successful as prevalence rates in these data are much higher than in previous studies. Ineffective sampling would have the opposite effect.

How you maintain the quality of sample 

21. We do this by a) having 2 staff members supervise sample collection, b) ensuring that swabbing is done for 20 seconds, and c) placing the sample on ice for immediate transfer to the laboratory. The high S. aureus prevalence in our data clearly show the success of our sampling. We added these points to the methods section and the discussion section. 

Statistical analysis is unclear 

22. Details on where the reviewer struggled to understand would help us address this concern. As noted above, we elaborated on the rigor of our statistical approach in the revised manuscript. We also added language describing the statistical limitations of using odds ratios to quantify the association between social relationships and colonization. 

Cite the Tables properly 

23. All Tables are now cited as per the PLOS ONE style guidelines. There were two instances where we wrote “see Table” – we deleted “see”.

Have you checked antimicrobial susceptibility profile of positives cases with regard to MRSA/VRSA 

24. We have not. There are many additional ways that we could characterize recovered strains, but this is beyond the scope of this work. See also our response below.

Have authors recommended decolonization of positive cases 

25. We did not, because a) this isn’t a clinical trial, b) these are otherwise healthy community members, and c) the assays are for research only and not approved for diagnostic testing. Decolonization (of MRSA) is for high risk patient populations not otherwise healthy community members and our ethical approval and funding did not encompass active surveillance testing for decolonization purposes. Decolonization is well outside the ethical and legal authority of our study. 

Status of MRSA and VRSA and co-resistance profiles have to be included 

26. We disagree for a number of reasons. First, while drug resistant strains will influence treatment of infections, there is no reason to believe that they impact carriage differently from drug sensitive strains. Second, the prevalence of drug resistant strains is very low and parsing out the data in this way would not be diagnostic (given at most a handful of MRSA isolates). Third, drug resistant strains have evolved independently multiple times from drug sensitive strains. Therefore, two MRSA strains may be more epidemiologically different from each other than a drug resistant and drug sensitive strain. 

Results

Please pinpoint the most relevant results rather than incurious findings 

27. Each of social relationship assessments has strong prognostic value for clinical endpoints, with the possible exception of social network size. Thus, our study includes the most potent social relationship resources and they are described in turn without emphasizing one over the other. We are not sure what an “incurious finding” is, unless this alludes to the null findings we report, which are encouraged by PLOS ONE policy. 

Discussion

Need [sic] substantial improvement. 

Please improve the main conclusion of the manuscript. 

28. We clearly state in the last paragraph that “structural social relationships and loneliness are not associated with colonization of S. aureus.” Because no specifics were provided for these comments, we did not further edit the discussion except for the addition of a new study (published after our initial submission) that also looks at the association of structural social relationships and loneliness with hospital-treated infection. This very large study found a small association of loneliness, but not social isolation, with hospital-treated infection. Looking more closely, the loneliness association was not observed when considering skin and soft tissue infections, which are mostly likely driven by S. aureus. Our data align with this new study, and provide a bridge between clinical infection and colonization. 

Reviewer #2: 1. The study appears to be well-conducted, but I recommend updating the references section to include more recent research on the topic. This will provide a more complete and current understanding of the subject and will also help to strengthen the overall credibility of the study. 

29. We are grateful to Reviewer 2 for the compliment. We assume that the reviewer is referring to our citations of estimates of prevalence in the US. Unfortunately there have not been more recent estimates, and the work that we cite is the most comprehensive estimate to date. We do however cite our previous work based in Arizona. No studies have addressed the connection between social network resources and carriage of S. aureus – our work is completely novel. As noted above, after we submitted our study, a study was published examining social relationships and infectious disease. That paper also notes the paucity of research in the area, and the only recent work on infection and social ties is related to COVID. We are aware of no other studies looking at colonization and social relationships, and if you search PubMed for these terms the only paper identified is an earlier paper from our group. 

2. The results of the manuscript appear to be weak and it would be beneficial to update the study with more recent findings. It is also important to highlight the use of whole genome sequencing in the results section, as it is a significant aspect of the study that should be emphasized and discussed in relation to the findings. This will help to strengthen the overall validity and impact of the study. 

30. We are not sure what the reviewer means by a “weak” result. We assume that he/she is referring to our inability to reject our null hypothesis? PLOS One clearly has an inclusive scope where “negative and null results are all in scope.“ Despite this, the implications of our work for mitigation practices are profound as we can conclude that casual social relationships are not likely to result in transmission. This work is very recent and represents our most recent findings. As noted above there is a paucity of published work on social relationships and infectious diseases, specifically pathogen carriage. We would be grateful for specific related references that are more recent but our literature search efforts do not uncover any.

Our whole genome sequencing results are only peripherally relevant to this work. The sequencing was done to validate our detection assay and is detailed elsewhere. What is relevant to this work is the sensitivity and specificity of our detection assay, and that has been detailed in 2 other publications that are cited. The excellent classification accuracy of our laboratory procedures indeed strengthens the validity of our work. Conversely, since this work is about carriage and social relationship resources, genome sequencing per se does not play a role and its inclusion would be a distraction.

There were other miscellaneous comments in a MS Word copy of the manuscript returned to the authors in the editorial decision email. These comments have been addressed above and in the manuscript (e.g., the Figure caption, which is now called a STROBE flow diagram as per the STROBE guidelines (cf. PLOS ONE 14(10): e0222359); elaborating on the self-swabbing p.10 ).

Sincerely,

Steven D. Barger PhD

Professor of Psychology

Steven.Barger@nau.edu

Department of Psychology

Northern Arizona University

(928) 523-9619 tel

(928) 523-6777 fax

---

## [Decision Letter · Decision Letter 1]

21 Mar 2023

PONE-D-23-00421R1Cross-sectional study of the association of social relationship resources with Staphylococcus aureus colonization in naturally occurring social groupsPLOS ONE

Dear Dr. Barger,

Thank you for submitting your manuscript to PLOS ONE. After careful consideration, we feel that it has merit but does not fully meet PLOS ONE’s publication criteria as it currently stands. Therefore, we invite you to submit a revised version of the manuscript that addresses the points raised during the review process.

Please incorporate the following minor corrections so that the manuscript would be acceptable for publication.

1. First person pronouns have been used frequently, the manuscript should be written anonymously and use of first person must be very limited.

2. Inclusion and exclusion criteria has to be precisely mentioned. Please submit your revised manuscript by May 05 2023 11:59PM. If you will need more time than this to complete your revisions, please reply to this message or contact the journal office at plosone@plos.org. Please include the following items when submitting your revised manuscript:A rebuttal letter that responds to each point raised by the academic editor and reviewer(s). You should upload this letter as a separate file labeled 'Response to Reviewers'.A marked-up copy of your manuscript that highlights changes made to the original version. You should upload this as a separate file labeled 'Revised Manuscript with Track Changes'.An unmarked version of your revised paper without tracked changes. You should upload this as a separate file labeled 'Manuscript'.If applicable, we recommend that you deposit your laboratory protocols in protocols.io to enhance the reproducibility of your results. Protocols.io assigns your protocol its own identifier (DOI) so that it can be cited independently in the future. For instructions see: https://journals.plos.org/plosone/s/submission-guidelines#loc-laboratory-protocols. Additionally, PLOS ONE offers an option for publishing peer-reviewed Lab Protocol articles, which describe protocols hosted on protocols.io. Read more information on sharing protocols at https://plos.org/protocols?utm_medium=editorial-email&utm_source=authorletters&utm_campaign=protocols.

We look forward to receiving your revised manuscript.

Kind regards,

Dario Ummarino, PhD

Senior Editor

PLOS ONE

On behalf of,

Dr. Samiullah Khan

Academic Editor

PLOS ONE

Journal Requirements:

Additional Editor Comments:

Dear Author,

Incorporate the following minor corrections so that the manuscript would be acceptable for publication.

1. First person pronouns have been used frequently, the manuscript should be written anonymously and use of first person must be very limited.

2. Inclusion and exclusion criteria has to be precisely mentioned.

Reviewers' comments:

Reviewer's Responses to Questions

**Comments to the Author**

1. If the authors have adequately addressed your comments raised in a previous round of review and you feel that this manuscript is now acceptable for publication, you may indicate that here to bypass the “Comments to the Author” section, enter your conflict of interest statement in the “Confidential to Editor” section, and submit your "Accept" recommendation.

Reviewer #1: (No Response)

Reviewer #2: All comments have been addressed

2. Is the manuscript technically sound, and do the data support the conclusions?

Reviewer #1: Partly

Reviewer #2: Yes

3. Has the statistical analysis been performed appropriately and rigorously? 

Reviewer #1: I Don't Know

Reviewer #2: Yes

4. Have the authors made all data underlying the findings in their manuscript fully available?

Reviewer #1: No

Reviewer #2: Yes

5. Is the manuscript presented in an intelligible fashion and written in standard English?

Reviewer #1: Yes

Reviewer #2: Yes

6. Review Comments to the Author

Reviewer #1: (No Response)

Reviewer #2: Dear Authors,

I have reviewed your manuscript titled "Cross-sectional study of the association of social relationship resources with Staphylococcus aureus colonization in naturally occurring social groups" and I am pleased to say that I found it to be a well-written and informative piece of research.

Your study sheds light on an important topic, and I believe that the findings will provide essential information for the scientific community. The results are presented in a clear and concise manner, and the methodology is sound. I particularly appreciated the thorough analysis of the data and the discussion of the potential implications of the results.

Overall, I commend you on your work and thank you for your contribution to the field. I wish you all the best in your future research endeavors.

Sincerely,

Dr. Umer

7. PLOS authors have the option to publish the peer review history of their article (what does this mean?). If published, this will include your full peer review and any attached files.

Reviewer #1: No

Reviewer #2: No

---

## [Author Response · Author response to Decision Letter 1]

24 Mar 2023

Please incorporate the following minor corrections so that the manuscript would be acceptable for publication. 

1. First person pronouns have been used frequently, the manuscript should be written anonymously and use of first person must be very limited. 

All first person language has been removed.

2. Inclusion and exclusion criteria has to be precisely mentioned. 

We elaborated on the inclusion and exclusion criteria in the Method section on p. 8.

---

## [Editor Report · Decision Letter 2]

29 Mar 2023

Cross-sectional study of the association of social relationship resources with Staphylococcus aureus colonization in naturally occurring social groups along the US/Mexico border

PONE-D-23-00421R2

Dear Dr. Steven Barger,

We’re pleased to inform you that your manuscript has been judged scientifically suitable for publication and will be formally accepted for publication once it meets all outstanding technical requirements.

Kind regards,

Samiullah Khan, Ph. D

Academic Editor

PLOS ONE
---

## [Editor Report · Acceptance letter]

4 Apr 2023

PONE-D-23-00421R2 

Cross-sectional study of the association of social relationship resources with *Staphylococcus aureus* colonization in naturally occurring social groups along the US/Mexico border 

Dear Dr. Barger:

I'm pleased to inform you that your manuscript has been deemed suitable for publication in PLOS ONE. Congratulations! Your manuscript is now with our production department. 

Kind regards, 

on behalf of

Dr. Samiullah Khan 

Academic Editor

PLOS ONE